# Influences of Aquaponics System on Growth Performance, Antioxidant Parameters, Stress Parameters and Gene Expression of *Carassius auratus*

Hanping Mao [1,*], Bin Wang [1], Jian Zhao [2], Yafei Wang [1], Xiaoxue Du [1] and Qiang Shi [3]

[1] School of Agricultural Engineering, Jiangsu University, Zhenjiang 212013, China; 2112016004@stmail.ujs.edu.cn (B.W.); 2111916018@stmail.ujs.edu.cn (Y.W.); 2111916004@stmail.ujs.edu.cn (X.D.)

[2] College of Biosystems Engineering and Food Science, Zhejiang University, Hangzhou 310000, China; zhaojzju@zju.edu.cn

[3] School of Science and Technology, Shanghai Open University, Shanghai 200433, China; shiqiang@sou.edu.cn

* Correspondence: maohpujs@163.com; Tel.: +86-135-1169-5868

**Abstract:** Aquaponics is a new type of composite farming system, which combines aquaculture and hydroponics through ecological design to achieve scientific synergism. However, the effects of aquaponics on the welfare status and stress parameters of fish are unclear. In this study, 150 crucian carp with an average initial body weight of $7.06 \pm 0.32$ g were selected. Nine fish were randomly selected as the control group (NC group and hypoxia group) for stress, antioxidant and gene expression parameters after acclimation and hypoxia stress, respectively. The remaining crucian carps were randomly divided into three experimental groups with 20 crucian carps in each group ($T_0$, $T_1$ and $T_2$, respectively), with three replicates. The fresh weight of the lettuce root in the $T_2$ group had no significant effect ($p > 0.05$). Compared with the control group ($T_0$ and $T_1$), there were significant differences in the specific growth rate, the weight gain rate, the fresh leaf weight, the chlorophyll content and the water quality parameters in group $T_2$ ($p < 0.05$). Regarding the biochemical parameters, superoxide dismutase and catalase showed significant differences between the $T_2$ and $T_1$ groups ($p < 0.05$). At the same time, the *HSP70* and *Prdx3* genes were upregulated in the liver of the $T_2$ group compared to the NC group and the hypoxia group. The research suggests that aquaponics may reduce the hypoxia stress of crucian carp without affecting the growth of crucian carp and lettuce.

**Keywords:** crucian carp; stress; antioxidant; aquaponics

**Key Contribution:** The symbiotic relationship between crucian carp and lettuce can ensure the growth parameters of lettuce and improve the living environment of crucian carp. We have demonstrated through growth parameters, physiological parameters and gene expression profiles that the fish–vegetable symbiosis model may alleviate the stress of crucian carp to a certain extent.

## 1. Introduction

Aquaponics is a compound culture system and a recent variation on the RAS systems. It is considered a promising and emerging system that combines intensive production with water conservation [1]. The aquaculture wastewater in the complex aquaculture system is not discharged to the outside. Insoluble large-particle feces produced by the fish are separated by physical filtration. The nitrifying bacteria are attached to the biochemical filtration system, and soluble wastes (such as ammonia nitrogen) are converted into nitrates through the nitrification of the nitrifying bacteria, which are then absorbed and used by the plants, while the purified water is returned to the fish pond [2]. When the system is stable, fish, microbes and aquatic plants can achieve a harmonious symbiotic relationship [3,4].

There are many important environmental factors affecting fish growth in an aquaponics system, such as the water temperature and oxygen [5,6]. An unsuitable water environment can cause fish to produce a stress response [7,8]. The aquaponics model may alleviate this problem to some extent. At present, the aquatic plants in the hydroponic system are mostly green leafy vegetables, such as lettuce [9], spinach [1], sweet pepper [2] and so on. Therefore, the aquaponics model has been vigorously developed. Some researchers have shown that aquatic plants can play certain roles in water quality regulation [10,11], such as increasing the oxygen content and cooling, shading and avoiding light.

The selection of fish in the aquaponics system is a key factor to determine the success of the aquaculture system [12]. The fish in the aquaponics system must be able to adapt to the turbidity of the water body and assimilate microbial proteins to a certain extent [13]. Crucian carp has excellent tolerance and strong adaptability to various ecological environments [14]. They have better growth performance and higher survival rates under different water quality conditions [15]. Therefore, they are considered to be one of the most suitable fish species for intensive farming.

The aquaponics system has many benefits for fish feeding, such as enhancing the growth performance and antioxidant capacity of fish [16,17]. Even under conditions of intensive culture, crucian carps have obvious advantages in terms of their antioxidant capacity and growth in aquaponics systems [15]. At the cellular level, their stress response is characterized by the increased expression of the heat shock protein (HSP) [18]. The increased synthesis of HSP70 has been reported in response to various stressors, including hypoxia, pathogens and contaminants [18,19]. Recombinant peroxiredoxin 3 (Prdx3) is closely related to antioxidant defense and tissue repair [20]. Therefore, HSP70 and Prdx3 can partly reflect the stress level of fish. This is conducive to the sustainable development of aquaculture to evaluate whether the aquaponics system can relieve the stress state of crucian carp while giving consideration to the high-quality growth of aquatic plants and fish. Currently, lettuce is mostly used in hydroponic systems and produces good results [21–23].

It is worth mentioning that under appropriate proportions, the fecal residue produced by fish in the aquaponics system will not affect the growth status of green leafy plants [24]. At the same time, we did not use any fertilizers or chemical plant protectors during the exploration process.

Therefore, lettuce was selected as the research object in this study. The study was only a preliminary pilot study. The purpose of this study was to explore the effects of an aquaponics system on the growth performance of crucian carp and lettuce quality, and to evaluate whether the aquaponics system has the ability to relieve crucian carp under oxygen stress.

## 2. Materials and Methods

### 2.1. Test Design and Installation

The crucian carps used in this study (150 crucian carps, $7.06 \pm 0.32$ g per culture unit) were obtained from Guangdong Xiongfeng Fry Co., Ltd., Shunde, China. They were initially acclimatized in RAS (recirculating aquaculture systems) for 30 days. The temperature of the water was held at 22–25 °C, the pH was held at 6.5–7.0, the ammonia nitrogen levels were held at 0–0.12 mg/L and the nitrite content was held at less than 0.12 mg/L. A 10% proportion of the water was replaced every day.

In the experiment, 150 crucian carps were initially included in the RAS. After domestication, 3 crucian carps were randomly selected as the NC group to detect stress, antioxidant parameters and liver gene expression, which was repeated three times. Then, the remaining 141 crucian carps were treated with hypoxia, and 3 crucian carps were randomly selected as the hypoxia group to detect stress, antioxidant parameters and liver gene expression, which was also repeated three times. As shown in Figure 1, after oxygen recovery, 120 crucian carps were randomly selected from 132 crucian carps and put into the water–fish group ($T_1$) and fish–vegetable group ($T_2$), with three replicates per treatment. There were 20 crucian

carps in each group. Among them, the crucian carps were first subjected to hypoxia stress and then put into $T_1$ and $T_2$ groups, respectively. The volume of each experimental tank was 240 L and the surface area of the lettuce tank was 60 cm$^2$. The stocking density of each tank was approximately 0.59 kg/m$^3$ As shown in Figure 2, the hydroponics unit in the test was composed of a filter box (a), biochemical filter cotton (b), plant bed (c), water inlet pipe (d), aquaculture tank (e), suction pump (f), lettuce (g), floating foam board (h), air pump (i), crucian carps (j) and gas stone (k). The plants were fixed using floating foam plates as a medium [2]. Pipes made of PVC were installed between the fishpond and the plant bed for circulating water. The bottom of the box at the top of the plant bed was lined with a double layer of biochemical filter cotton. This allowed only filtered water to enter the plant bed, preventing damage to plant roots that can be caused by high concentrations of suspended matter in circulating water, which can cause adhesion or blockage, finally reducing the ability of the roots to absorb oxygen. The water was filtered through the submersible pump to the top filter box, and then it circulated back to the plant bed due to gravity, and ultimately back to the fishpond. Laboratory staff adjusted the water level in the tank every week to compensate for losses due to evaporation, transpiration, handling losses and so on.

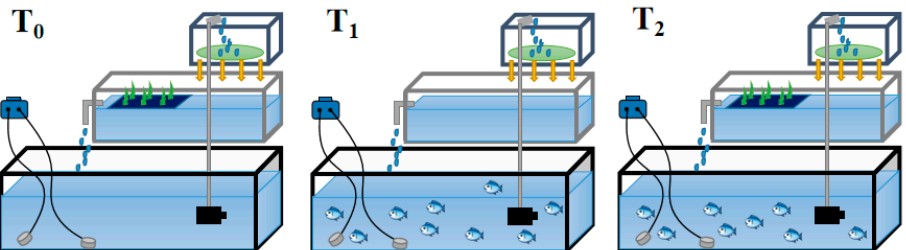

**Figure 1.** The three experimental groups: ($T_0$) water–vegetable group; ($T_1$) water–fish group; ($T_2$) fish–vegetable group.

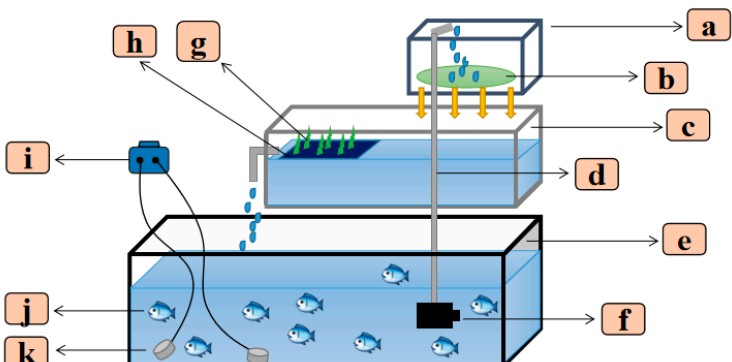

**Figure 2.** Hydroponics unit in the test: (**a**) filter box; (**b**) biochemical filter cotton; (**c**) plant bed; (**d**) water inlet pipe; (**e**) aquaculture tank; (**f**) suction pump; (**g**) lettuce; (**h**) floating foam board; (**i**) air pump; (**j**) crucian carp; (**k**) air stone.

### 2.2. Treatment of Test Fish

During this period, the fish were placed in a 12 h/12 h light–dark cycle and artificially fed twice a day with commercial pellets (crude protein 35.0%, crude fiber 12.0%, crude fat 5.0%, crude ash 15.0%, total phosphorus 1.0%, moisture 12.5% and lysine 1.6%) (Shandong Binzhou Ruixing Biotechnology Co., Ltd., Binzhou, China) until they felt full. The maximum value of the daily feed was 2% of the total body weight of the fish. The water quality parameters were measured once a day. Finally, 120 hypoxic crucian carps were taken out quickly and put into the $T_1$ and $T_2$ groups for 30 days, respectively. At the end of the experiment, the growth parameters of the surviving experimental fish were first measured,

then 3 fish from each group were randomly selected for sampling, and this process was conducted three times. The parameter abbreviations are shown in Table 1.

**Table 1.** Parameter Abbreviations.

| Parameter | Parameter Abbreviation |
| --- | --- |
| SGR | Specific growth rate |
| WGR | Relative growth rate |
| SR | Survival rate |
| SOD | Total superoxide dismutase |
| CAT | Catalase |

At the end of the trial, all fish were collected to obtain the final weight and the following growth indices were calculated [2]:

$$\text{Specific growth rate (SGR)} = 100\% \times (\ln W_2 - \ln W_1)/\text{days}, \tag{1}$$

$$\text{Relative growth rate (WGR)} = 100\% \times (W_2 - W_1)/W_1, \tag{2}$$

$$\text{Survival rate (SR)} = 100\% \times (N_2/N_1) \tag{3}$$

Here, $W_1$ is the initial weight, $W_2$ is the final weight, $N_2$ is the final number and $N_1$ is the initial number.

*2.3. Lettuce Growth Indices*

The lettuce seeds were cultured in hole trays for 7 days using soil as the medium. After germination, the lettuce was separated from the hole dish and the roots were removed from the soil by washing and transferred to a floating foam board with stationary cotton. The average height of the plants was $4.16 \pm 0.12$ cm, and there were 6 plants per floating foam plate, which remained unchanged in all treatments and controls. At the end of the 30-day trial period, the heights of all plants were measured using vernier calipers, then the samples were washed and the leaves and roots of the plants were weighed using an electronic balance with a minimum measuring range of 0.01 g after wiping the water. Chlorophyll content was indirectly reflected by the SPAD value (relative chlorophyll content), which was measured using a portable chlorophyll analyzer (SPAD-502; KONICA MINOLTA, Osaka, Japan).

*2.4. Stress Parameters*

Experimental crucian carp were randomly selected from each pond and each fish was sampled. All samples were taken after anesthetizing them with MS-222. Growth performance data were obtained and then blood was drawn from the tail vein with heparinized syringes and centrifuged at $1350\times g$ for 10 min. The serum was collected 5 h after coagulation at 4 °C and used for the determination of glucose. At the same time, we collected the livers of the crucian carp and turned them into tissue homogenate to measure the cortisol and antioxidant parameters later on.

The activity levels of total superoxide dismutase and catalase were measured. The total superoxide dismutase activity was measured using the hydroxylamine method at 550 nm. The activity of the catalase was determined at 405 nm using the ammonium molybdate method [25]. The cortisol content was determined with the competition method at 450 nm [26]. The kit was acquired from Nanjing Jiancheng Bioengineering Institute (Nanjing, China). The glucose level was determined with a Cobas C-311 automatic biochemical analyzer (Roche, Basel, Switzerland). The results of the stress parameters and antioxidant parameters were expressed as means $\pm$ SDs.

### 2.5. Extraction and Quantitative PCR of Tissue RNA

The total RNA of the tissues was isolated using the TRIzol reagent (Invitrogen, California, CA, USA). The reverse transcription was performed using the HiScript 1st Strand cDNA Synthesis kit (Vazyme, Piscataway, NJ, USA), and 1000 ng of total RNA was used in this process. The reverse transcription was carried out at 37 °C for 15 min and 85 °C for 5 s. The qPCR reactions were performed using the Universal SYBR qPCR kit (Vazyme, Piscataway, NJ, USA). The qRT-PCR analysis of the mRNA was carried out with a QuantStudio$^{TM}$ 3 Real-Time PCR Instrument (Thermo Fisher Scientific, Massachusetts, USA). The thermal cycling parameters were as follows: 95 °C for 5 min; 40 cycles at 95 °C for 15 s, 60 °C for 30 s and 60–95 °C to draw the dissociation curve. The expression level of the mRNA was normalized to the expression of GAPDH. The $2^{-\Delta\Delta Ct}$ method was used for quantification. The primer sequences are shown in Table 2.

**Table 2.** Primer sequences.

| Gene | Forward Primer | Reverse Primer |
|---|---|---|
| *HSP70* | 5′–ACTGAACTCGGTCATTGGCT–3′ | 5′–AGAGGCCAATTGCAGTTCAT–3′ |
| *Prdx3* | 5′–TCGCAGTCTCAGTGGATTCC–3′ | 5′–CAGGAGGCATTGCTGATGAT–3′ |
| *GAPDH* | 5′–CAGGAGGCATTGCTGATGAT–3′ | 5′–GAAGGCTGGGGCTCATTT–3′ |

### 2.6. Statistical Analysis

The data were analyzed using GraphPad Prism 5.0 software (GraphPad Software). The growth performance, growth, stress and antioxidant parameters of the crucian carps were analyzed using a one-way ANOVA. Here, $p < 0.05$ indicates a statistically significant difference. The results are expressed as means ± SDs.

## 3. Results

### 3.1. Fish Growth Performance

At the end of the trial, the survival rate was 100% in both the $T_1$ and $T_2$ groups. The growth properties of the crucian carp are shown in Table 3. The FW, SGR, WGR and SR values were significantly different between the $T_1$ and $T_2$ groups ($p < 0.05$), as shown in Table 1. The FW, SGR and WGR values were higher in the $T_2$ group, at 10.26 ± 0.31, 1.25 ± 0.10 and 45.51 ± 4.43, respectively. The FW, SGR and WGR values in the $T_1$ group were lower, at 9.12 ± 0.40, 0.89 ± 0.05 and 30.63 ± 2.14, respectively.

**Table 3.** Growth properties of crucian carp in $T_1$ and $T_2$ groups.

| Parameter | $W_1$ (g) | $W_2$ (g) | SGR (%) | WGR (%) | SR (%) |
|---|---|---|---|---|---|
| $T_1$ | 7.00 ± 0.40 [a] | 9.12 ± 0.40 [b] | 0.89 ± 0.05 [b] | 30.63 ± 2.14 [b] | 100 [a] |
| $T_2$ | 7.11 ± 0.24 [a] | 10.26 ± 0.31 [a] | 1.25 ± 0.10 [a] | 45.51 ± 4.43 [a] | 100 [a] |

Results are expressed as means ± SDs ($n = 9$) and different letters indicate significant differences ($p < 0.05$).

### 3.2. Plant Growth Parameters

The fresh leaf weight, root fresh weight and chlorophyll content of the lettuce are shown in Table 4. There were significant differences in the leaf fresh weight and chlorophyll content between the $T_0$ group and $T_2$ group ($p < 0.05$); the $T_2$ group had a higher leaf fresh weight. The chlorophyll content in the $T_2$ group was higher than that in the $T_0$ group. There was no significant difference in the root fresh weights between the two groups, and the root fresh weight in the $T_2$ group was higher than that in the $T_0$ group.

**Table 4.** Leaf fresh weight, root fresh weight and chlorophyll contents of lettuces in $T_0$ and $T_2$ groups.

| Parameter | Leaf Fresh Weight (g plant$^{-1}$) | Fresh Root Weight (g plant$^{-1}$) | Chlorophyll (SPAD) |
|---|---|---|---|
| $T_0$ | 32.80 ± 0.50 [b] | 6.93 ± 0.42 [a] | 31.64 ± 0.78 [b] |
| $T_2$ | 34.33 ± 0.65 [a] | 7.64 ± 0.36 [a] | 34.40 ± 0.43 [a] |

Results are expressed as means ± SDs (*n* = 9), and different letters indicate significant differences (*p* < 0.05).

### 3.3. Water Quality Parameters

The water quality parameters of groups $T_0$, $T_1$ and $T_2$ are shown in Table 5. There were significant differences in the pH, dissolved oxygen, nitrate and nitrite levels among the three groups during the whole experiment (*p* < 0.05). The pH values ranged from 6.9 to 7.8, and the levels of dissolved oxygen ranged from 5.35 to 7.21 mgL$^{-1}$. The pH of the $T_1$ group was the highest, followed by the $T_2$ group, while the $T_0$ group was the lowest. The highest level of dissolved oxygen was found in the $T_0$ group, followed by the $T_2$ group and the $T_1$ group. The nitrate level in the $T_1$ group was the highest, followed by group $T_2$, and the $T_0$ group was the lowest. The nitrite levels ranged from 0.06 to 0.27 mgL$^{-1}$, with the highest nitrite level being found in the $T_1$ group, followed by the $T_2$ group, with the lowest in the $T_0$ group.

**Table 5.** Study of the water quality parameters under the conditions of groups $T_0$, $T_1$ and $T_2$.

| Parameter | pH | Dissolved Oxygen (mg/L) | Nitrate (mg/L) | Nitrite (mg/L) |
|---|---|---|---|---|
| $T_0$ | 7.05 ± 0.03 [c] | 7.01 ± 0.05 [a] | 3.40 ± 0.13 [c] | 0.08 ± 0.01 [c] |
| $T_1$ | 7.52 ± 0.04 [a] | 5.56 ± 0.10 [c] | 6.83 ± 0.10 [a] | 0.24 ± 0.01 [a] |
| $T_2$ | 7.19 ± 0.01 [b] | 6.71 ± 0.03 [b] | 5.19 ± 0.09 [b] | 0.12 ± 0.01 [b] |

Results are expressed as means ± SDs (*n* = 9), and different letters indicate significant differences (*p* < 0.05).

### 3.4. Stress Parameters

The stress parameters (cortisol and glucose) of crucian carp are shown in Figure 3. The difference in cortisol levels was statistically significant (*p* < 0.05); the cortisol concentrations in the hypoxia, $T_1$, and $T_2$ groups were significantly higher (*p* < 0.05) compared to the NC group. The group with the highest cortisol concentration was the hypoxia group, followed by groups $T_1$ and $T_2$ in that order. The glucose concentration in the hypoxia group was significantly higher than that in the $T_1$ group, while in the $T_2$ group, it was slightly lower than that in $T_1$ group. Furthermore, there was no significant difference in glucose concentrations between the $T_2$ and NC groups.

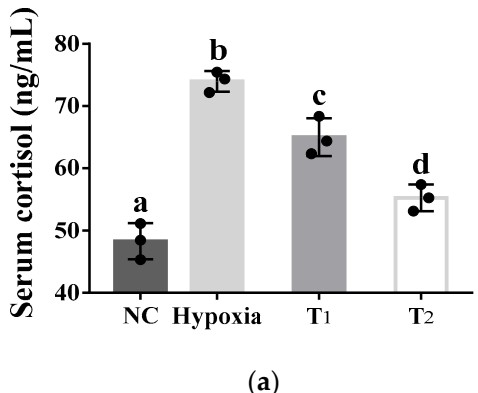

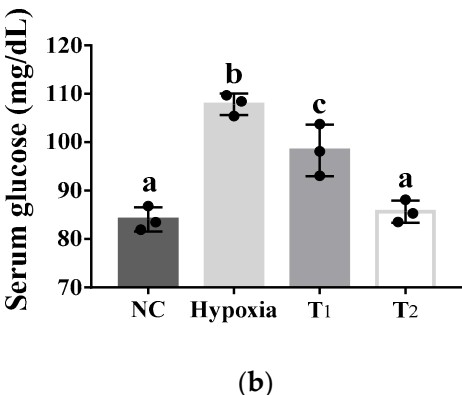

(a)　　　　　　　　　　　　　　　　　(b)

**Figure 3.** Stress parameters of crucian carp in NC, hypoxia, $T_1$ and $T_2$ groups: (**a**) cortisol; (**b**) serum glucose. Results are expressed as means ± SDs (*n* = 9), and different letters indicate significant differences (*p* < 0.05).

### 3.5. Antioxidant Parameters

The antioxidant parameters of crucian carp are shown in Figure 4, and the differences in the catalase and superoxide dismutase activities were statistically significant ($p < 0.05$). Compared to the NC group, the CAT concentration was significantly higher ($p < 0.05$) in the hypoxia, $T_1$ and $T_2$ groups. The hypoxia group had the highest CAT concentration, followed by the $T_1$ group, while the $T_2$ group had the lowest CAT concentration. The SOD concentration in the hypoxia group was significantly higher than that in the $T_1$ group, whereas the SOD concentration in the $T_2$ group was significantly lower than that in the $T_1$ group. Additionally, the SOD concentration in the $T_2$ group did not differ significantly from that in the NC group.

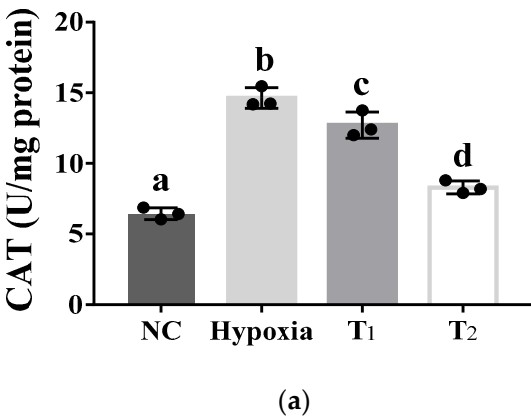
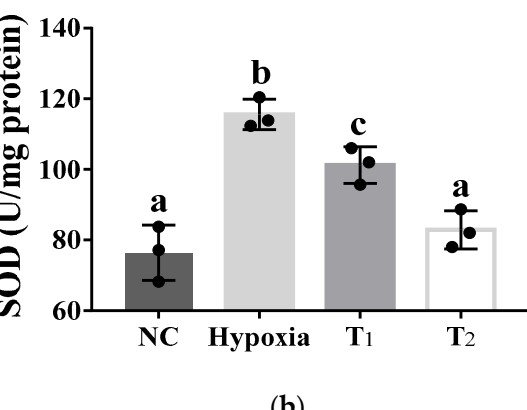

(**a**)

(**b**)

**Figure 4.** Antioxidant parameters of crucian carp in the NC, hypoxia, $T_1$ and $T_2$ groups were studied: (**a**) CAT; (**b**) SOD. Results are expressed as means ± SDs ($n = 9$), and different letters indicate significant differences ($p < 0.05$).

### 3.6. Gene Expression

The relative expression levels of *HSP70* and *Prdx3* in crucian carp liver tissues are shown in Figure 5. The relative expression of *HSP70* in the control group was the lowest ($p < 0.05$), while the relative expression of *HSP70* in the hypoxia group was significantly higher than that in the $T_1$ group, followed by the $T_2$ group ($p < 0.05$). On the other hand, compared with the other three groups, the relative expression of *Prdx3* in healthy crucian carp in the control group was significantly higher ($p < 0.05$), and the relative expression of *Prdx3* in the $T_2$ group was significantly higher than that of the $T_1$ group, followed by the hypoxia group ($p < 0.05$).

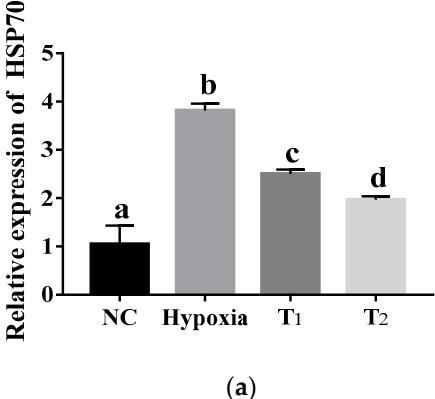
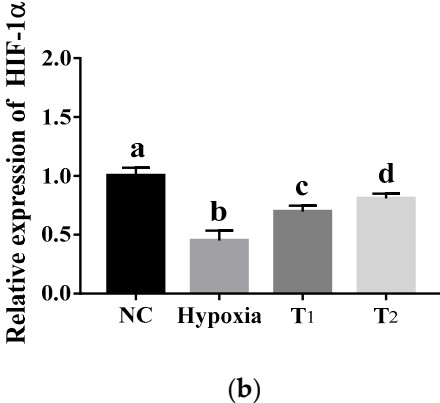

(**a**)

(**b**)

**Figure 5.** Average relative expression of hypoxia stress-related genes in the livers of crucian carp under different treatment conditions: (**a**) *HSP70*; (**b**) *Prdx3*. Results are expressed as means ± SDs ($n = 9$), and different letters indicate significant differences ($p < 0.05$).

## 4. Discussion

With the development of aquaculture technology, aquaponics models have been adopted by more and more researchers for related research. However, the information on the effect of this model on the stress level of crucian carp is lacking. The effects of the aquaponics model on the stress of crucian carp were studied in terms of the growth parameters, water quality parameters, antioxidant parameters, stress parameters and gene expression.

In this experiment, the FW, SGR and WGR values of the $T_2$ group were higher than those of the $T_1$ group, which may have been because the aquatic plants [27] improved the living environment conditions of the crucian carp and promoted their growth performance [9]. Some studies have shown that hydroponic systems may contribute to fish growth [28]. In addition, this experiment further proved that under certain density conditions, the water body purified by aquatic plants may improve the water environment, which may improve the growth performance of the fish. At the same time, the survival rate of crucian carp may be guaranteed to a certain extent [29].

Aquaponics may ensure the normal growth of leafy greens [30,31]. This may be due to the fact that higher quantities of nitrate and nitrite in the water [32] are absorbed and utilized by aquatic plants. It has been reported that nitrate and nitrite can promote the vegetative growth of aquatic plants [33,34]. The chlorophyll content is one of the physiological indexes used to evaluate the nutritional status of plants [35]. The chlorophyll content of plants in the $T_2$ group was significantly higher. This may indicate that the photosynthetic capacity of the plants in the aquaponics system was slightly better than those in the traditional hydroponic system. The aquaponics system may have improved the availability of the nitrogen in the form of nitrate and enabled the root system to better absorb nitrogen, thereby improving the photosynthetic capacity [36,37].

Lettuce grown in hydroponics may be involved in regulating water quality through its own photosynthesis. The pH of the $T_1$ group was significantly high. The excrement from the crucian carp in group T1 was probably the reason for the alkalinity of the water and the higher pH. In the $T_2$ group, due to the absorption of nitrate in the water by lettuce, the water remained weakly alkaline [38]. The level of dissolved oxygen in the $T_1$ and $T_2$ groups was significantly lower than that in the $T_0$ group because of the oxygen consumption of the crucian carp ($p < 0.05$). The level of dissolved oxygen in the $T_2$ group was higher than that in the $T_1$ group, possibly because the lettuce photosynthesis provided dissolved oxygen in the water [39]. The $T_2$ group had low levels of nitrate and nitrite, possibly due to the absorption of these two components by the plants [40].

Cortisol and glucose are important parameters to measure whether fish are in a state of stress [41]. The cortisol and glucose levels of the crucian carp in the $T_2$ group were significantly lower than those in the $T_1$ group. Good water quality may reduce the adverse irritation to fish. Therefore, the crucian carp were less affected by stress in group $T_2$.

The antioxidant parameters of fish can be reflected by the activities of superoxide dismutase and catalase, and the greater the activity change, the more serious the damage to the fish [42]. In organisms, superoxide dismutase plays an important role in antioxidant defense mechanisms by participating in the conversion of superoxides to oxygen and hydrogen peroxide [43]. In addition, catalase also plays an important role in defense mechanisms by breaking down hydrogen peroxide into oxygen and water [44]. In this study, the levels of catalase and superoxide dismutase in the crucian carp in the $T_1$ group were significantly increased, while those in the $T_2$ group were lower. This may have been due to the existence of antioxidant components in plant roots, which may reduce the oxidative stress of fish to some extent [45]. In this study, the levels of superoxide dismutase and catalase were lower, indicating that the stress levels and health status of the crucian carp were better in the symbiotic system.

*HSP* plays a significant role in protecting cells from protein damage and promoting cell growth [20]. In this study, the expression of *HSP70* was significantly increased in the hypoxia group ($p < 0.05$), indicating that the liver can protect cell homeostasis by

upregulating *HSP70* gene expression [46]. When the dissolved oxygen in the $T_1$ and $T_2$ groups returned to normal levels, the *HSP70* gene expression recovered to the pre-stress level, which was slightly higher than that of the NC group. *Prdx3* is believed to be crucial in the process of antioxidant defense and tissue repair [20]. The hypoxia group significantly reduced the expression level of *Prdx3*, suggesting that the liver can improve the defense system and accelerate the repair of damaged tissues by downregulating the gene expression of *Prdx3*. In this study, it was observed that after the $T_1$ and $T_2$ groups returned to normal dissolved oxygen levels, the gene expression of *Prdx3* returned to near pre-stress levels, slightly lower than that of the NC group.

## 5. Conclusions

This was a pilot study. The addition of aquatic plants may alleviate the stress of crucian carp. The model of aquaponics under the experimental conditions produced good results on the related parameters of the crucian carp and lettuce. In addition, by analyzing physiological parameters (cortisol, glucose, superoxide dismutase, catalase) and non-invasive parameters such as the chlorophyll content of the aquatic plants, we can judge whether the overall effects on fish and aquatic plants have been improved. This process is informative and cost-effective. At the same time, this experiment shows that the symbiotic model can alleviate the stress state of fish to a certain extent.

**Author Contributions:** Conceptualization, B.W. and H.M.; methodology, B.W., J.Z., Q.S. and Y.W.; software, B.W., J.Z. and Y.W.; validation, B.W., Q.S. and X.D.; data curation, B.W. and J.Z.; writing—original draft preparation, B.W. and X.D.; writing—review and editing, B.W., Y.W. and J.Z.; project administration, H.M.; funding acquisition, H.M. All authors have read and agreed to the published version of the manuscript.

**Funding:** This research was funded by the National Natural Science Foundation of China (32071905), National Natural Science Foundation of China (NSFC) (32201686), the Project of Agricultural Equipment Department of Jiangsu University (NZXB20210106), and the Modern Agriculture Industrial Technology System Special Project—the National Technology System for Conventional Freshwater Fish Industries (grant no. CARS-45-26).

**Institutional Review Board Statement:** The maintenance, handling and experiments conducted on the fish during this study were carried out in strict accordance with the guidelines of the Association for the Study of Animal Behavior of Zhejiang University (no. ZJU20190073).

**Informed Consent Statement:** Not applicable.

**Data Availability Statement:** Data available on request due to restrictions. The data presented in this study are available on request from the corresponding author. The data are not publicly available due to privacy reasons.

**Acknowledgments:** The authors would like to thank the Key Laboratory of Agricultural Engineering in Jiangsu University and Zhejiang University for supporting the experimental conditions of the research.

**Conflicts of Interest:** The authors declare no conflict of interest.

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
