# Peer review of "Influences of Aquaponics System on Growth Performance, Antioxidant Parameters, Stress Parameters and Gene Expression of Carassius auratus"

_fishes, doi:10.3390/fishes8070360_

Round 1
Reviewer 1 Report
The MS reports important findings about the integrated carp-lettuce aquaponics system. As explained by the authors, the study is a pilot but the results obtained suggest that this integrated aquaponic system may reduce the hypoxia stress of fis without affecting their growth and that of the plant.
The MS is worthy of interest and reports significant outcomes. In my opinion, there are only moderate revisions that should be carried on before the MS could be considered for publication. They are all given in the pdf attached.
To summarize:
- Introduction: some sentences could be moved to another part. Moreover, I would suggest adding some sentences (paragraphs) to underline the reasons why this fish species was selected.
- methods: some sentences should be rephrased in terms of English. They are not clear or the English terminology is not the more adequated for a scientific paper
- results: some of the figures could be changed to table to better explain the results obtained (see the pdf for details)
- discussion: avoid the use of sentences that are not well supported by references. The relationship between the findings and the reason for this results can be only a supposition/suggestion and should be considered in this way without further certian experiments.
- conclusion: the authors should underline here again that this is a pilot study and its novelty.

The MS could benefit of a check by an English fluently speaker.
Author Response
1.Review expert recommendations
Comments and Suggestions for Authors
The MS reports important findings about the integrated carp-lettuce aquaponics system. As explained by the authors, the study is a pilot but the results obtained suggest that this integrated aquaponic system may reduce the hypoxia stress of fis without affecting their growth and that of the plant.
The MS is worthy of interest and reports significant outcomes. In my opinion, there are only moderate revisions that should be carried on before the MS could be considered for publication. They are all given in the pdf attached.
To summarize:
- Introduction: some sentences could be moved to another part. Moreover, I would suggest adding some sentences (paragraphs) to underline the reasons why this fish species was selected.
- methods: some sentences should be rephrased in terms of English. They are not clear or the English terminology is not the more adequated for a scientific paper.
- results: some of the figures could be changed to table to better explain the results obtained (see the pdf for details)
- discussion: avoid the use of sentences that are not well supported by references. The relationship between the findings and the reason for this results can be only a supposition/suggestion and should be considered in this way without further certian experiments.
- conclusion: the authors should underline here again that this is a pilot study and its novelty.
---------------------------------------------------------------------------------------------------------------------------------
1.Respond to the experts
(1)Introduction: some sentences could be moved to another part. Moreover, I would suggest adding some sentences (paragraphs) to underline the reasons why this fish species was selected.
Responses: We are very sorry for causing you trouble. According to your comments, we have added some sentences "Crucian carp has excellent tolerance and strong adaptability to various ecological environments. They have better growth performance and higher survival rates under different water quality conditions. Therefore, they are considered to be one of the most suitable fish for intensive farming." in lines 60-66.
In addition, our previous experiments have proved that crucian carp is highly adaptable to the environment and can be applied to aquaponics systems.
We are very sorry for causing you trouble. According to your comments, we have put the "Due to the wide variety of green leafy vegetables suitable for aquaponics models, it has a certain universality. " change to " Therefore, the model of aquaponics has been vigorously developed " in lines 53-54. We have put the sentence "Therefore, lettuce was selected as the research object in this study." in the last paragraph of the introduction in line 83.
(2)methods: some sentences should be rephrased in terms of English. They are not clear or the English terminology is not the more adequated for a scientific paper.
Responses: Thanks for your suggestion. According to your comment, We have rephrased the corresponding sentences in the manuscript. Thank you for your valuable advice.
(3)results: some of the figures could be changed to table to better explain the results obtained (see the pdf for details).
Responses: We are very sorry for causing you trouble. According to your comments, we have converted Figure 3 into a table on lines 218-220. Meanwhile we have changed the "(a)" to black in line 230.
- discussion: avoid the use of sentences that are not well supported by references. The relationship between the findings and the reason for this results can be only a supposition/suggestion and should be considered in this way without further certian experiments.
Responses: We are very sorry for causing you trouble. According to your comments, we have added the summary paragraph "With the rise of aquaculture technology, aquaponics model has been adopted by more and more researchers for related research. However, the effect of this model on stress level of crucian carp is lacking. The effects of aquaponics model on stress of crucian carp were studied in terms of growth parameters, water quality parameters, antioxidant parameters, stress parameters and gene expression." in lines 264-268.
We have put the sentence "The excrement of crucian carp in group T1 made the water body alkaline, so its pH was the highest." into" The excrements of crucian carp in group T1 were probably the reason for the alkalinity of the water, and the higher pH." in lines 294-296.
(5)conclusion: the authors should underline here again that this is a pilot study and its novelty.
Responses: We are very sorry for causing you trouble. According to your comments, We have changed the sentence "it is a better choice to establish the symbiotic mode under the conditions of this experiment" to "The model of aquaponics under the experimental conditions produced good results on the related parameters of crucian carp and lettuce" in lines 328-329.
We have added "This is a pilot study. The addition of aquatic plants may alleviate the stress of crucian carp." in the conclusion in lines 327-328.
(6)According to your questions in the annotated pdf version, we will mark the remaining questions in red in the manuscript. Here are the details of the changes to the remaining issues.
We have changed "fish" to "specimens of crucian carp" in line 17. We have changed "crucian carp" to "fish" in line 17. We have replaced "improper" with "not suitable" in line 48.
We raised 150 test fish in our initial recirculating aquaculture system. First, we randomly selected three test fish each time as a control group, and measured stress parameters, antioxidant parameters and gene expression respectively. This part is carried out three times. Then, we subjected the remaining 141 test fish to hypoxia treatment. At this time, three fish were selected as the hypoxia group, and stress parameters, antioxidant parameters and gene expression were measured respectively. Finally, we divided the remaining 132 fish into three groups, with an average of 20 fish per group. The remaining 12 test fish have been kept in the original recirculating aquaculture system in lines 97-104.
We have removed "as shown in Figure 1" in line 111. We add references in lines 111 of the manuscript.
We have moved "The crucian carps used in this study (150 crucian carps, 7.06±0.32 g per culture unit) were obtained from Guangdong Xiongfeng Fry Co., LTD., Shunde, China. They were initially acclimatized in RAS (Recirculating Aquaculture Systems) for 30 days. Temperature of the water was held at 22-25℃, pH was held at 6.5-7.0, ammonia nitrogen levels were held at 0-0.12 mg/L and nitrite content was held at less than 0.12 mg/L. A 10% proportion of the water was replaced every day." to section 2.1" on lines 90-95.
We change the title of 2.2 from "Acquisition and Treatment of test fish" to "Treatment of test fish" in line 129.
At the end of the experiment, we also randomly selected 3 fish from 6 groups, including fish-water group and fish-vegetable group, and sampled them to determine stress parameters, antioxidant parameters and gene expression. This process was carried out 3 times, so n=9 represents this process. Before this, the growth parameters of all surviving crucian carp in each group were determined. Therefore, we repeat the sentence "At the end of the experiment, the growth parameters of the surviving experimental fish are first measured. Then, each group randomly selects 3 fish for sampling processing, and this process is conducted three times." in lines 137-140.
The references we have added are in lines 155 and 435-437. We have changed "Three crucian carps" to "Experimental Crucian Carp" in line 157. We have removed "followed by the T0 group" in line 206.
We have added new references "Matteo Zarantoniello,Giulia Chemello,Stefano Ratti,Lina Fernanda Pulido-Rodríguez,Enrico Daniso,Lorenzo Freddi,Pietro Salinetti,Ancuta Nartea,Leonardo Bruni,Giuliana Parisi,Paola Riolo,Ike Olivotto.Growth and Welfare Status of Giant Freshwater Prawn (Macrobrachium rosenbergii) Post-Larvae Reared in Aquaponic Systems and Fed Diets including Enriched Black Soldier Fly (Hermetia illucens) Prepupae Meal. Animals 2023, 13(4), 715." on lines 453-456.
Reviewer 2 Report
Peer review
Evaluation and opinion about the article of: Hanping Mao, Bin Wang, Jian Zhao, Yafei Wang, Xiaoxue Du, Qiang Shi: Growth and physiological performance of Crucian carp in a carp-lettuce aquaponics system
· Need to write out the scientific names of the species involved in the experiment at the first appearance.
· Need to find a more accurate title for the article
· The abstract is not accurate. Need to introduce aquaponics, because this is not discussed in the chapter.
· Insert an abbreviation list into the text.
· Please in the introduction chapter write a little bit more about the interaction of the crucian carp and the aquaponics systems. I think we need to know how the fish species has been chosen. What were the points of view that you used the crucian carp?
· Line 92: Please describe exactly the stocking density (total biomass weight/m3) of the experiment based on an m3.
· Line 124: Was the protein level optimal? The 5% of protein was very low. Can we get appropriate results with such a feed like that?
· Line 182: what kind of program was used for the statistical analysis?
· Table 1. Technically there was no weight gain during the experiment what is understandable if we check the diet formulation. Why the authors think that such a performances are relevant in any relations?
· In Table 1. the smaller value signed with letter “a”, but at the Table 2. the bigger one. Please unify the values!
· The labeling of the significant differences are confusing at Figure 3., Figure 4., Figure 5., Figure 6. Please make the corrections.
· Line 268: The authors write that aquatic plants can improve the growth performance of fish in a high density. I am not convinced that we can say that the experiment has run at high density, and because the survival rate was 100% in every treatment we cannot say that the hydroponics conditions have a positive effect on the survival rate.
· The statement that the plant biomass weight is more in the tank where fish can be found is obvious, because of the nutrient contents.
· The chapter from line 270-286 about the plant performance was too general, like almost the whole discussion part.
My overall opinion, that after major revision, especially the discussion part, the article can be accepted for publication.
none
Author Response
2.Respond to the experts
Peer review
Evaluation and opinion about the article of: Hanping Mao, Bin Wang, Jian Zhao, Yafei Wang, Xiaoxue Du, Qiang Shi: Growth and physiological performance of Crucian carp in a carp-lettuce aquaponics system
Need to write out the scientific names of the species involved in the experiment at the first appearance.
Need to find a more accurate title for the article.
The abstract is not accurate. Need to introduce aquaponics, because this is not discussed in the chapter.
Insert an abbreviation list into the text.
Please in the introduction chapter write a little bit more about the interaction of the crucian carp and the aquaponics systems. I think we need to know how the fish species has been chosen. What were the points of view that you used the crucian carp?
Line 92: Please describe exactly the stocking density (total biomass weight/m3) of the experiment based on an m3.
Line 124: Was the protein level optimal? The 5% of protein was very low. Can we get appropriate results with such a feed like that?
Line 182: what kind of program was used for the statistical analysis?
Table 1. Technically there was no weight gain during the experiment what is understandable if we check the diet formulation. Why the authors think that such a performances are relevant in any relations?
In Table 1. the smaller value signed with letter “a”, but at the Table 2. the bigger one. Please unify the values!
The labeling of the significant differences are confusing at Figure 3., Figure 4., Figure 5., Figure 6. Please make the corrections.
Line 268: The authors write that aquatic plants can improve the growth performance of fish in a high density. I am not convinced that we can say that the experiment has run at high density, and because the survival rate was 100% in every treatment we cannot say that the hydroponics conditions have a positive effect on the survival rate.
The statement that the plant biomass weight is more in the tank where fish can be found is obvious, because of the nutrient contents.
The chapter from line 270-286 about the plant performance was too general, like almost the whole discussion part.
My overall opinion, that after major revision, especially the discussion part, the article can be accepted for publication.
---------------------------------------------------------------------------------------------------------------------------------
(1)Need to write out the scientific names of the species involved in the experiment at the first appearance.
Responses: We are very sorry for causing you trouble. According to your comments, we have replaced the place where the test animals first appeared with the scientific name of the species "Carassius auratus" in lines 2-4.
(2)Need to find a more accurate title for the article.
Responses: Thanks for your suggestion. According to your comment, we have changed the title of the manuscript to "Effects of Aquaponics System on Growth Performance, Antioxidant Parameters, Stress Parameters and Gene Expression of Carassius auratus" in lines 2-4.
(3)The abstract is not accurate. Need to introduce aquaponics, because this is not discussed in the chapter.
Responses: We are very sorry for causing you trouble. According to your comments, we have added this sentence at the beginning of the summary "Aquaponics is a new type of composite farming system, which combines aquaculture and hydroponics through ecological design to achieve scientific synergism. However, the effects of aquaponics on the welfare status and stress parameters of fish are unclear." in lines 14-16.
(4)Insert an abbreviation list into the text.
Responses: We are very sorry for causing you trouble. According to your comments, we have added a list of abbreviations in line 189.
(5)Please in the introduction chapter write a little bit more about the interaction of the crucian carp and the aquaponics systems. I think we need to know how the fish species has been chosen. What were the points of view that you used the crucian carp?
Responses: We are very sorry for causing you trouble. According to your comments, we have added some sentences in lines 60-66. Crucian carp has excellent tolerance and strong adaptability to various ecological environments. They have better growth performance and higher survival rates under different water quality conditions. Therefore, they are considered to be one of the most suitable fish for intensive farming.
In addition, our previous experiments has proved that crucian carp is highly adaptable to the environment and can be applied to aquaponics systems.
(6)Line 92: Please describe exactly the stocking density (total biomass weight/m3) of the experiment based on an m3.
Responses: We are very sorry for causing you trouble. According to your comments, we have provided supplementary description of the stocking density of each test tank in lines 107-108.
(7)Line 124: Was the protein level optimal? The 5% of protein was very low. Can we get appropriate results with such a feed like that?
Responses: We are very sorry for causing you trouble. According to your comments, we wrote wrong parameters of commercial particles. We have checked and corrected the parameters of commercial particles in lines 131-133.
(8)Line 182: what kind of program was used for the statistical analysis?
Responses: We are very sorry for causing you trouble. Data analysis was performed using GraphPad Prism 5.0 software. Differences between groups were calculated by multiple comparison test by one-way ANOVA.
(9)Table 1. Technically there was no weight gain during the experiment what is understandable if we check the diet formulation. Why the authors think that such a performances are relevant in any relations?
Responses: We are very sorry for causing you trouble. According to your comments, we have carefully checked the formula of the feed pellets and the protein content is 35% instead of 5%, which we have corrected in lines 131-133.
(10)In Table 1. the smaller value signed with letter “a”, but at the Table 2. the bigger one. Please unify the values!
Responses: We are very sorry for causing you trouble. According to your comments, we have unified the letters in the table.
(11)The labeling of the significant differences are confusing at Figure 3., Figure 4., Figure 5., Figure 6. Please make the corrections.
Responses: We are very sorry for causing you trouble. According to your comments, we have corrected the labeling of significant differences in figures 3, 4 and 5.
(12)Line 268: The authors write that aquatic plants can improve the growth performance of fish in a high density. I am not convinced that we can say that the experiment has run at high density, and because the survival rate was 100% in every treatment we cannot say that the hydroponics conditions have a positive effect on the survival rate.
Responses: We are very sorry for causing you trouble. According to your comments, we have revised this sentence to“In addition, this experiment further proves that under certain density conditions, the water body purified by aquatic plants may improve the water environment, which may improve the growth performance of fish. At the same time, the survival rate of crucian carp may be guaranteed to a certain extent.”on lines 273-276.
(13)The chapter from line 270-286 about the plant performance was too general, like almost the whole discussion part.
Responses: We are very sorry for causing you trouble. According to your comments, we have made changes to the discussion in lines 264-301.
Round 2
Reviewer 2 Report
There are some self citations in the text. My opinion that is not the luckiest choice to make such a self citations (9,10,18,29). I suggest to remove it from the reference list and of course from the text too.
Author Response
1.Comments and Suggestions for Authors
There are some self citations in the text. My opinion that is not the luckiest choice to make such a self citations (9,10,18,29). I suggest to remove it from the reference list and of course from the text too.
1.Respond to the experts
(1)There are some self citations in the text. My opinion that is not the luckiest choice to make such a self citations (9,10,18,29). I suggest to remove it from the reference list and of course from the text too.
Responses: We are very sorry for causing you trouble. According to your comments, we have removed references (9, 10, 18, 29) from the manuscript. At the same time, we have also removed from the literature reference list.